# Inhibition of chitin deacetylases to attenuate plant fungal diseases

Lin Liu [1,2,3,9], Yeqiang Xia[4,5,6,9], Yingchen Li[3,9], Yong Zhou[7], Xiaofeng Su[8], Xiaojing Yan[3], Yan Wang [4,5,6], Wende Liu [3], Hongmei Cheng[8] ✉, Yuanchao Wang [4,5,6] ✉ & Qing Yang [1,2,3] ✉

Phytopathogenic fungi secrete chitin deacetylase (CDA) to escape the host's immunological defense during infection. Here, we showed that the deacetylation activity of CDA toward chitin is essential for fungal virulence. Five crystal structures of two representative and phylogenetically distant phytopathogenic fungal CDAs, *Vd*PDA1 from *Verticillium dahliae* and Pst_13661 from *Puccinia striiformis* f. sp. *tritici*, were obtained in ligand-free and inhibitor-bound forms. These structures suggested that both CDAs have an identical substrate-binding pocket and an Asp-His-His triad for coordinating a transition metal ion. Based on the structural identities, four compounds with a benzo-hydroxamic acid (BHA) moiety were obtained as phytopathogenic fungal CDA inhibitors. BHA exhibited high effectiveness in attenuating fungal diseases in wheat, soybean, and cotton. Our findings revealed that phytopathogenic fungal CDAs share common structural features, and provided BHA as a lead compound for the design of CDA inhibitors aimed at attenuating crop fungal diseases.

Plant diseases cause serious economic losses and threaten food accessibility in many areas[1,2]. To date, more than 10,000 phytopathogenic fungi, including soil-borne and air-borne fungi, have been identified to cause various plant diseases that currently cannot be effectively controlled[3,4].

In the battle between plant hosts and phytopathogenic fungi, plants first recognize microbe-associated molecular patterns (MAMPs) via cell-surface receptors, which elicit immune responses[5]. Chitin, a β1,4-linked *N*-acetyl-glucosamine biopolymer found in fungal cell walls, is a well-known MAMP[6]. Plant hosts secrete chitinase (EC 3.2.1.14) to degrade chitin and release chitooligosaccharides (COs)[7–10], which are sensed by specific plant lysin motif-containing cell-surface receptors[11]. Recognition by these receptors activates intracellular immune signaling via mechanisms such as mitogen-associated protein kinases (MAPKs) pathways, ultimately preventing fungal colonization[12–15]. However, recent research has indicated that pathogenic fungi can avoid this defensive mechanism by secreting a chitin-remodeling enzyme known as chitin deacetylase (CDA) (EC 3.5.1.41) to deacetylate chitin[16–18]. The resulting deacetylated chitin, termed chitosan[19,20], is a poor substrate for chitinases[19,20]. Gao et al.[16] showed that the cotton root pathogen *Verticillium dahliae* secretes a CDA (*Vd*PDA1) that is specifically localized to fungus-host interfaces. *Vd*PDA1 prevents CO-induced immune response reactive oxygen species (ROS) production, MAPK phosphorylation, and resistance-related gene

[1]School of Bioengineering, Dalian University of Technology, 116024 Dalian, China. [2]Shenzhen Branch, Guangdong Laboratory of Lingnan Modern Agriculture, Key Laboratory of Synthetic Biology, Ministry of Agriculture and Rural Affairs, Agricultural Genomics Institute at Shenzhen, Chinese Academy of Agricultural Sciences, 518000 Shenzhen, China. [3]State Key Laboratory for Biology of Plant Diseases and Insect Pests, Institute of Plant Protection, Chinese Academy of Agricultural Sciences, 100193 Beijing, China. [4]Department of Plant Pathology, Nanjing Agricultural University, 210095 Nanjing, China. [5]Key Laboratory of Soybean Disease and Pest Control (Ministry of Agriculture and Rural Affairs), Nanjing Agricultural University, 210095 Nanjing, China. [6]The Key Laboratory of Plant Immunity, Nanjing Agricultural University, 210095 Nanjing, China. [7]School of Software, Dalian University of Technology, 116024 Dalian, China. [8]Biotechnology Research Institute, Chinese Academy of Agricultural Sciences, Beijing 100081, China. [9]These authors contributed equally: Lin Liu, Yeqiang Xia, Yingchen Li. ✉e-mail: chenghongmei@caas.cn; wangyc@njau.edu.cn; qingyang@caas.cn

expression. Similarly, *Fusarium oxysporum* f. sp. *vasinfectum* secretes a CDA (*Fov*PDA1) to enhance its virulence, causing cotton wilt disease[16]. Xu et al.[18] reported that the wheat leaf-infecting pathogen *Puccinia striiformis* f. sp. *tritici* secretes a CDA (Pst_13661) during the early stages of infection. Transgenic wheat plants carrying *Pst_13661* siRNA were resistant to *P. striiformis* f. sp. *tritici* and displayed reduced ROS, downregulated immune gene expression levels, and delayed development of disease symptoms. Similar mechanisms involving CDAs have been proposed for other pathogenic fungi, such as the powdery mildew fungus *Blumeria graminis*[21], broad bean rust fungus *Uromyces fabae*[10], maize anthracnose fungus *Colletotrichum graminicola*[22], rice blast fungus *Pyricularia oryzae*[23], and corn smut fungus *Ustilago maydis*[24].

CDAs are metalloenzymes belonging to the carbohydrate esterase family 4 (CE4) (www.cazy.org), which also includes peptidoglycan deacetylases, MurNAc deacetylases, poly-β1,6-GlcNAc deacetylases, and acetylxylan esterases[25–29]. All CDAs contain a conserved catalytic domain known as the NodB homology domain, which includes a conserved Asp-His-His triad that coordinates metal ions for activity[26,30–32]. A common metal-assisted acid/base mechanism was proposed for three fungal CDAs with known structures: *Cl*CDA from *Colletotrichum lindemuthianum*[33], *An*CDA from *Aspergillus nidulans* FGSC A4[34], and *Ang*CDA from *Aspergillus niger*[32]. A water molecule (coordinated by a divalent metal cation) is deprotonated by the catalytic base, aspartic acid, which acts as a nucleophile that attacks carbonyl carbon in the substrate, producing a tetrahedral oxyanion intermediate. The nitrogen group of this intermediate is then protonated by the catalytic acid histidine, resulting in the release of an acetate product. To date, there are no known structures of any phytopathogenic fungal CDAs, hampering efforts to target these enzymes.

In this study, we determined the crystal structures and inhibitors of *Vd*PDA1 and Pst_13661, which are derived from soil-borne and air-borne phytopathogenic fungal CDAs, respectively, and represent two major sources of agricultural pathogenic fungi. We obtained benzohydroxamic acid (BHA), a lead compound, by structure-guided screening, which showed high activity against diseases in cotton, wheat, and soybean caused by *V. dahliae* and four other top 10 fungal plant pathogens for molecular plant pathology, namely *P. striiformis* f. sp. *tritici*, *F. oxysporum*, *Fusarium graminearum*, and *Rhizoctonia solani*. This study not only provides the structural features of phytopathogenic fungal CDAs but also identifies CDA inhibitors that are promising for controlling plant diseases.

## Results
### Structures of *Vd*PDA1 and Pst_13661
*Vd*PDA1 and Pst_13661 are single-domain enzymes with 55% sequence similarity. These two enzymes were cloned and recombinantly produced in yeast. The purified enzymes were then crystallized using the hanging-drop vapor-diffusion method, and the crystal structures were obtained at resolutions of 2.64 Å and 1.96 Å, respectively, via molecular replacement with *Cl*CDA as the searching model (Table 1).

The substrate-binding grooves of *Vd*PDA1 and Pst_13661 are short and open-ended (Fig. 1a, b). Both enzymes adopt a classic $(\beta/\alpha)_7$ fold. The active site of *Vd*PDA1, which is in the center of the substrate-binding groove, superimposes well onto that of Pst_13661, with a root mean square deviation of 0.43 Å over 32 Cα atoms, suggesting that these two enzymes have a structurally identical substrate-binding pocket.

In the structure of *Vd*PDA1, the signature Asp-His-His triad (formed by Asp56, His108 and His112) and a water molecule were

**Table 1 | Data collection and refinement statistics**

|  | *Vd*PDA1 apo[#] | Pst_13661 apo | Pst_13661- BHA | Pst_13661-compound 2 | Pst_13661-compound 3 |
|---|---|---|---|---|---|
| **Data collection** |  |  |  |  |  |
| Space group | P2₁2₁2 | P2₁2₁2₁ | P2₁2₁2₁ | P2₁2₁2₁ | P2₁2₁2₁ |
| Cell dimensions |  |  |  |  |  |
| *a, b, c* (Å) | 138.617, 125.899, 127.007 | 39.669, 65.727, 105.362 | 39.786, 65.420, 106.031 | 39.859, 65.528, 105.966 | 39.754, 65.693, 106.806 |
| α, β, γ (°) | 90.00, 90.00, 90.00 | 90.00, 90.00, 90.00 | 90.00, 90.00, 90.00 | 90.00, 90.00, 90.00 | 90.00, 90.00, 90.00 |
| Resolution (Å) | 2.64–50.00 (2.64–2.70)* | 1.96–50.00 (1.96–1.99) | 1.61–50.00 (1.61–1.64) | 1.61–50.00 (1.61–1.64) | 1.93–50.00 (1.93–1.96) |
| $R_{merge}$ | 0.144 (0.520) | 0.147 (0.413) | 0.134 (0.421) | 0.140 (0.442) | 0.173 (0.497) |
| *I/σI* | 14.7 (4.3) | 11.1 (3.1) | 16.8 (5.0) | 12.3 (3.8) | 10.0 (3.2) |
| Completeness (%) | 100 (100) | 99.3 (90.1) | 99.2 (95.2) | 97.8 (91.1) | 99.9 (100.0) |
| Redundancy | 13.2 (13.1) | 11.2 (7.4) | 13.0 (12.1) | 13.1 (11.1) | 12.4 (9.6) |
| **Refinement** |  |  |  |  |  |
| Resolution (Å) | 2.64–35.13 | 1.96–32.86 | 1.61–25 | 1.61–20.91 | 1.93–22.17 |
| No. reflections | 65535 | 20309 | 36370 | 35667 | 21675 |
| $R_{work}/R_{free}$ | 0.1924/0.2270 | 0.1947/0.2246 | 0.1646/0.1866 | 0.1573/0.1800 | 0.1729/0.1930 |
| No. atoms |  |  |  |  |  |
| Protein | 8545 | 1794 | 1794 | 1807 | 1800 |
| Ligand/ion | 5 | 1 | 11 | 25 | 15 |
| Water | 647 | 167 | 298 | 360 | 284 |
| *B*–factors |  |  |  |  |  |
| Protein | 24.77 | 27.67 | 16.54 | 13.72 | 13.87 |
| Ligand/ion | 51.00 | 67.21 | 25.98 | 37.78 | 38.91 |
| Water | 28.78 | 36.53 | 33.12 | 31.89 | 27.28 |
| R.m.s. deviations |  |  |  |  |  |
| Bond lengths (Å) | 0.008 | 0.010 | 0.007 | 0.006 | 0.013 |
| Bond angles (°) | 0.99 | 0.96 | 0.92 | 0.90 | 1.11 |

[#]one xtal for each structure. *Values in parentheses are for highest-resolution shell.

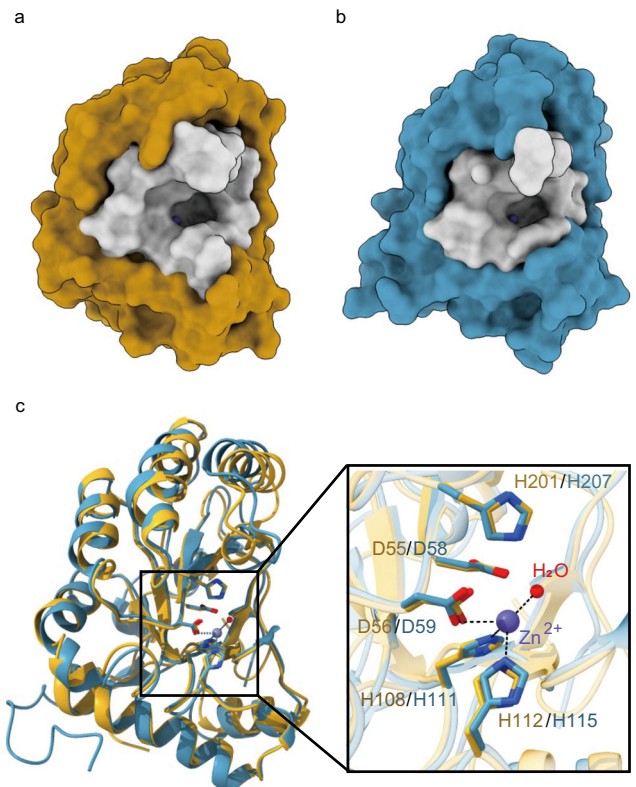

**Fig. 1 | Overall structures and active sites of *Vd*PDA1 and Pst_13661. a, b** Cartoon representation of the overall structures of *Vd*PDA1 (**a**) and Pst_13661 (**b**), and surface representation of their substrate-binding pockets (gray). The zinc ion is shown as a sphere in the slate. **c** Alignment of the overall structures of the two enzymes and surface representation of their active sites. *Vd*PDA1 is shown in orange and Pst_13661 is shown in blue. The metal-binding triad (Asp56/Asp59-His108/His111-His112/His115), catalytic residues (general base Asp55/Asp58 and general acid His201/His207), a Zn$^{2+}$ ion (slate sphere), and a water molecule (red sphere) are shown. The amino acids are labeled by their one-letter abbreviations.

coordinate with a metal ion (Fig. 1c). The X-ray fluorescent spectra (Supplementary Fig. 1a) showed that there were two peaks, one each for Ni$^{2+}$ and Zn$^{2+}$, respectively. To identify which metal ion was coordinated by the Asp-His-His triad in the active pocket, we performed zinc single-wavelength anomalous dispersion (SAD) of Pst_13661 (Supplementary Fig. 1b) and determined that zinc was the metal ion. Kinetic studies also indicated that both *Vd*PDA1 and Pst_13661 used the Zn$^{2+}$ ion instead of the Ni$^{2+}$ ion for their higher activity (Supplementary Fig. 1c, d). In this scenario, Ni$^{2+}$ was likely to be a contaminant sequestered from nickel-nitrilotriacetic acid (Ni-NTA) affinity chromatography during purification. Based on the catalytic mechanisms of CE4 deacetylases[32], Asp55 and His201 of *Vd*PDA1 were presumed to act as the catalytic base and catalytic acid, respectively.

## Enzymatic activity for virulence
The mutant *Vd*PDA1$^{D55A, H201A}$, which completely lost its deacetylase activity, was constructed (Fig. 2a). To determine whether CDA activity is essential for virulence, a *Vd*PDA1-deletion strain (Vd*Δpda1*) and complement strains with either mutated *Vd*PDA1$^{D55A, H201A}$ (Vd*Δpda1/cPDA1$^{DSSA, H201A}$*) or the wild-type *Vd*PDA1 (Vd*Δpda1/cPDA1*) were generated (Fig. 2b). Similar to the Vd*Δpda1*[16], both the Vd*Δpda1/cPDA1$^{DSSA,H201A}$* and Vd*Δpda1/cPDA1* strains grew normally (Fig. 2c), suggesting that CDA activity is not required for fungal growth in vitro. However, both Vd*Δpda1* and Vd*Δpda1/cPDA1$^{DSSA, H201A}$* strains caused significantly reduced cotton wilt symptoms and biomass (Fig. 2d, e). The expression levels of cotton (*Gossypium hirsutum* cv. Lvmian) mitogen-activated

protein kinase 6 (*Gh*MPK6) and respiratory burst oxidase homolog protein D (*Gh*RbohD) genes were much higher in plants inoculated with Vd*Δpda1* and Vd*Δpda1/cPDA1$^{DSSA, H201A}$* fungal strains compared to those inoculated with the wild-type strain (Fig. 2f, g). These two genes are chitin-induced defense response genes previously discovered to be present in the roots of cotton plants[16,35–37]. In contrast, Vd*Δpda1/cPDA1* retained its virulence, as evidenced by the increased biomass of *V. dahliae* and the decreased expression levels of *Gh*MPK6 and *Gh*RbohD (Fig. 2e–g), suggesting that *V. dahliae* relied on the enzymatic activity of *Vd*PDA1 for full virulence.

## BHA as a lead compound for designing CDA inhibitors
As the crystal structures revealed that Zn$^{2+}$ was coordinated by the Asp-His-His triad, we considered metal ion chelators, specifically BHA and its derivatives, as potential inhibitors of CDA. Four compounds, all of which contained a BHA moiety, showed good inhibitory activities toward *Vd*PDA1 and Pst_13661 (Table 2, Supplementary Fig. 2). BHA, the most potent inhibitor, had $K_i$ values of 8.31 μM and 9.83 μM toward *Vd*PDA1 and Pst_13661, respectively.

To confirm our hypothesis that CDA inhibitors could act as anti-plant disease agents, BHA was sprayed onto cotton leaf surfaces before fungi inoculation (Fig. 3a). With BHA treatment, the plants did not exhibit wilt symptoms (Fig. 3b), and the fungal biomass was significantly reduced in a BHA dose-dependent manner (Fig. 3c). Moreover, the colony morphology and hyphae growth of *V. dahliae* was not affected by BHA (Supplementary Fig. 3), indicating that BHA acted as a fungal virulence attenuator, instead of as a fungicide.

To determine whether BHA could recover the defensive responses of infected host plants, the expression levels of *Gh*MPK6 and *Gh*RbohD in the roots of cotton plants were analyzed after infection with the wild type, the knockout mutant (Vd*Δpda1*), and the complementary transformants (Vd*Δpda1/cPDA1$^{DSSA, H201A}$*, Vd*Δpda1/cPDA1*) in the presence or absence of BHA (Fig. 3d–f). In the absence of BHA, the wild type and Vd*Δpda1/cPDA1* suppressed *Gh*MPK6 and *Gh*RbohD expression; however, Vd*Δpda1* and Vd*Δpda1/cPDA1$^{DSSA, H201A}$* upregulated *Gh*MPK6 and *Gh*RbohD expression (Fig. 3e, f, pink columns). In the presence of BHA, all strains upregulated *Gh*MPK6 and *Gh*RbohD expression (Fig. 3e, f, orange columns). These results suggest that BHA might inhibit CDA in vivo.

## BHA is a metal ion chelator of CDAs
To prove that BHA functioned as a metal ion chelator of CDAs, the crystal structures of Pst_13661 in complex with BHA and compounds **2** and **3** were resolved (Table 1). BHA chelates a zinc ion in bidentate mode (Fig. 4a). The hydroxamic acid moiety of BHA forms three hydrogen bond interactions with the enzyme: (i) its C=O with the NH backbone of Tyr152, (ii) its NH with the imidazole group of catalytic His207, and (iii) its OH with the carboxylic acid group of Asp58. The benzene ring moiety of BHA fits a cavity formed by the nonpolar residue Leu205 and the aromatic residues Tyr152 and Trp174. Similar to the inhibitory mechanism of BHA, the structures of Pst_13661 complexed with compounds **2** and **3** suggest that the hydroxamic acid moiety is important for inhibition as it chelates the catalytically important zinc ion and interacts with the catalytic residues (Fig. 4b, c).

## BHA demonstrates broad-spectrum activity
Phylogenetic analysis of fungal CE4 deacetylases indicated that *Vd*PDA1 homologs are widely distributed among plant-pathogenic fungi, such as *V. dahliae*, *P. striiformis* f. sp. *tritici*, *F. oxysporum*, *F. graminearum*, and *R. solani* (Fig. 5a). The catalytic center is highly conserved among these phylogenetically distant enzymes (Fig. 5b, c). To determine whether BHA is also effective against these enzymes, *Fo*PDA1 from *F. oxysporum*, *Fg*CDA from *F. graminearum*, and *Rs*CDA

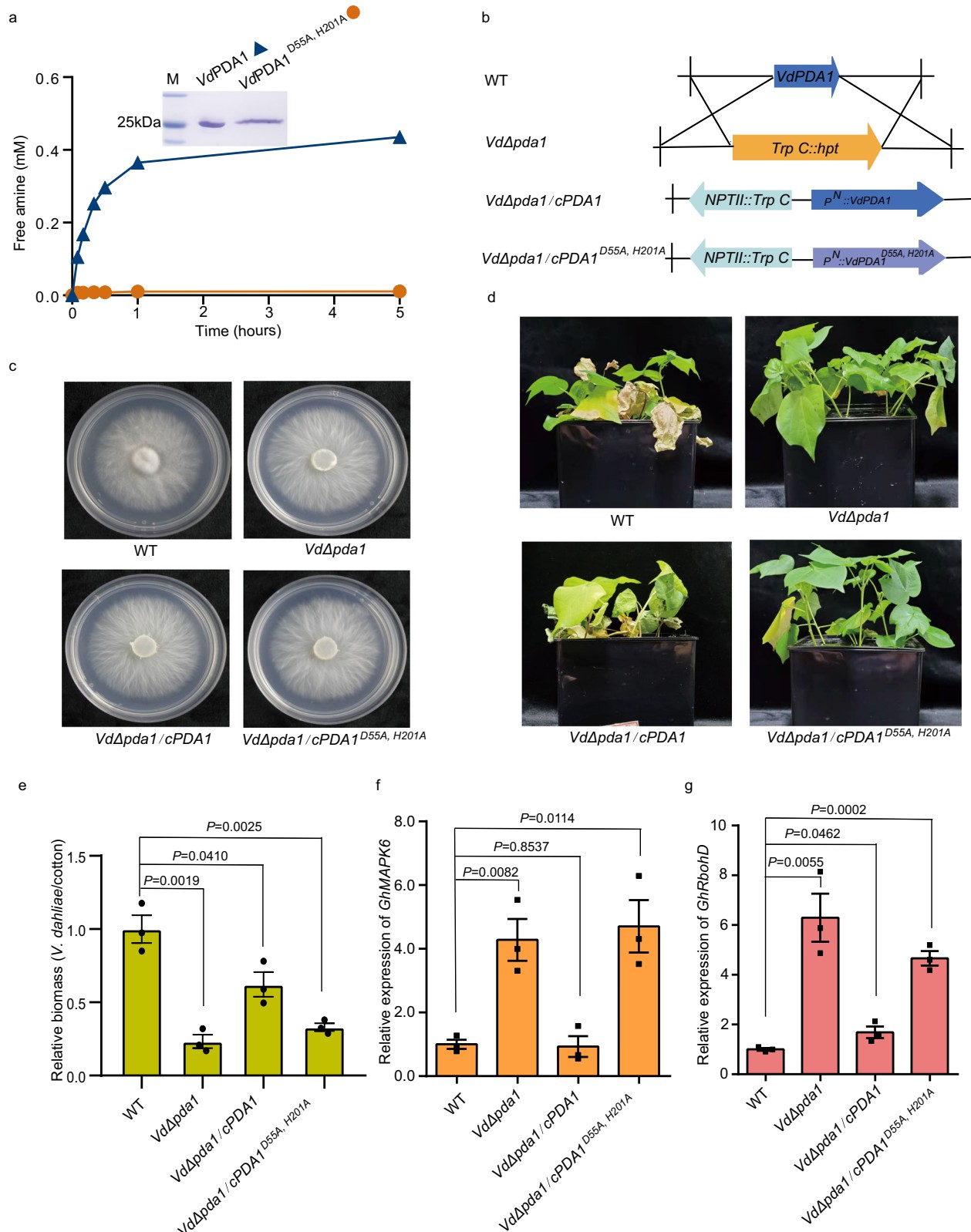

from *R. solani* were cloned and recombinantly expressed. Inhibition assays confirmed the broad-spectrum activity of BHA toward these enzymes, with $K_i$ values of 20.73 μM, 43.5 μM, and 2.39 μM for *Fg*CDA, *Fo*PDA1, and *Rs*CDA, respectively (Fig. 5d–f). Accordingly, the anti-disease activity of BHA was consistent with the inhibitory data. As shown in Fig. 5g, h, soybean hypocotyls infected with *F. graminearum*, *F. oxysporum*, or *R. solani* had fewer lesions and significantly

reduced fungal biomass in the presence of BHA. However, all BHA-treated fungi showed normal morphology and developmental status (Supplementary Fig. 3). Furthermore, disease indices after *P. striiformis* f. sp. *tritici* infection greatly declined with the application of various concentrations of BHA to wheat plants (Fig. 5i). These results demonstrate that BHA is effective in attenuating plant fungal diseases.

**Fig. 2 | Enzymatic activity of *Vd*PDA1 is essential for the full virulence of *V. dahliae*. a** Mutagenesis analysis confirmed the importance of the predicted catalytic residues for enzymatic activity. Coomassie blue-stained SDS-PAGE analysis of purified *Vd*PDA1 and *Vd*PDA1$^{D55A, H201A}$ (inset panel). M represents protein molecular weight marker. Time-dependent deacetylation of (GlcNAc)$_3$ by *Vd*PDA1 and *Vd*PDA1$^{D55A, H201A}$, determined via labeling the resulting free amine using fluorescamine and quantification using a glucosamine standard. All measurements were carried out in triplicate. **b** Schematic description of the *Vd*PDA1 mutants. *Trpc*: a promoter of *Aspergillus nidulans* trpC gene; *hpt*: hygromycin phosphotransferase; *NPTII*: neomycin phosphotransferase II; $P^N$: a native promoter of *Vd*PDA1. **c** There were no growth differences among *Vd*PDA1 mutant colonies grown on PDA medium. **d** Disease symptoms on cotton plants at 30 dpi with wild type (WT) or *Vd*PDA1 mutants. **e** Enzymatic activity is essential for the full virulence function of VdPAD1.

The fungal relative biomass on cotton plants at 30 dpi with the WT, the knockout mutant (VdΔ*pda1*), and the complementary transformants (VdΔ*pda1/cPDA1*$^{D55A, H201A}$, VdΔ*pda1/cPDA1*) was determined by qPCR, using genomic DNA as a template. **f, g** Expression of the defense response genes (*GhMPK6* and *GhRbohD*) that was repressed by VdPDA1 during infection by *V. dahliae*. The expression patterns of *GhMPK6* (**f**) and *GhRbohD* (**g**) in cotton roots at 36 hpi with the WT, the knockout mutant (VdΔ*pda1*), and the complementary transformants (VdΔ*pda1/cPDA1*$^{D55A, H201A}$, VdΔ*pda1/cPDA1*) were determined by qRT-PCR using cDNA as a template. Three independent replicates were used for measurement, and each experiment consisted of 3 cotton samples per *V. dahliae* infection. Bars in **e**–**g** represent mean values, whiskers represent mean ± SEM, and black circles represent each replicate value (*n* = 3). Statistical significance was determined by Student's two-sided unpaired *t*-test. Source data are provided as a Source Data file.

## Discussion

Mounting evidence suggests that fungal virulence factors can be potential targets for the development of anti-disease agents[12,38–40]. Fungal CDA, an effector essential for phytopathogenic fungal virulence, is highly promising in this regard. Using *Vd*PDA1 as a model, we showed that the catalytic residue-mutated strain VdΔ*pda1/cPDA1*$^{D55A, H201A}$ could not evade host plant defenses, suggesting that the deacetylase activity of fungal CDAs is crucial for full virulence (Fig. 2). Therefore, inhibiting phytopathogenic fungal CDAs is a promising strategy for controlling plant diseases.

In this study, BHA and compounds with a BHA moiety were discovered to inhibit the activity of phytopathogenic fungal CDAs, the enzymes that use chitin as substrate. Not surprising, BHA exhibits high potential for controlling the occurrence of crop diseases induced by phytopathogenic fungi. The crystal structures of Pst_13661 complexed with BHA or its derivatives suggest that BHA inhibits the activity of Pst_13661 by binding to the zinc ion. This is consistent with the fact that BHA is a chelator of metal ions and an inhibitor of certain metalloprotein enzymes, such as urease, histone deacetylases, and matrix metalloproteases[41–46]. We also showed that BHA and its derivatives could not inhibit insect chitin deacetylases (Supplementary Fig. 4), suggesting that BHA has a preference for fungal chitin deacetylases. These findings reveal opportunities for the design of more effective inhibitors that target pathogenic fungal CDAs by starting with the BHA scaffold.

Our results indicate that BHA does not affect the growth of pathogenic fungi, including *F. oxysporum*, *F. graminearum*, and *R. solani*, as well as the commonly used bio-agent fungus *Trichoderma harzianum* (Supplementary Fig. 3). Thus, BHA is more likely to be a plant immune activator and might not target beneficial fungi. However, it is possible that BHA might inhibit other metalloproteins that play a role in immune escape from the host plant's defensive responses. Further research is required to investigate other possible mechanisms involved.

## Methods
### Protein expression and purification

*V. dahliae* *Vd*PDA1 wild-type, *Vd*PDA1$^{D55A, H201A}$ (containing mutated catalytic residues), and *P. striiformis* f. sp. *tritici* Pst_13661 were gene synthesized and codon optimized for expression in yeast using Genscript (China). The DNA fragments were ligated into pPIC9 expression vectors. The resulting expression plasmids were subsequently linearized with the restriction enzyme Sal I to allow integration into the chromosomal DNA of *Pichia pastoris* GS115 (Catalog number C18100, Invitrogen, Carlsbad, CA, USA). Recombinant *P. pastoris* was first grown in buffered complex medium containing glycerol (BMGY, Invitrogen) at 301 K to an optical cell density of 4.0 at 600 nm. The cells were collected by centrifugation, resuspended in buffered methanol complex medium (BMMY, Invitrogen), and transferred into a 5 L fermentation tank. The culture volume used for the production of recombinant proteins was 3 L. The pH was controlled using a sterilized

base solution of 1 M KOH. Protein production was induced by adding methanol to the vessel at a constant feed rate. Fermentation proceeded for 92 h at 301 K. The culture supernatant was obtained by centrifugation.

The supernatant was subjected to ammonium sulfate precipitation with 75% saturation at 277 K for 12 h. After centrifugation, the supernatant was removed, and the precipitate was resuspended in distilled water and then desalted in buffer A (20 mM sodium phosphate, 0.5 M sodium chloride; pH 8.0) using a HiTrap Desalting column (5 ml; GE Healthcare, USA). The resulting sample was then loaded into a HisTrap HP affinity column (5 ml; GE Healthcare, USA) equilibrated in Buffer A. The target proteins were eluted with 20 mM sodium phosphate, 0.5 M NaCl, and 150 mM imidazole at pH 8.0. SDS-PAGE analysis showed that the eluted protein was > 95% pure. Uncropped scan of gel is provided as a Source Data file.

### Crystallization and data collection

The purified proteins were desalted in 20 mM Tris (pH 8.0) and 100 mM NaCl, and concentrated to 10 mg/mL for the crystallization experiments. The crystallization conditions were screened using hanging-drop vapor diffusion in 96-well VDX plates at 277 K. The drop consisted of equal volumes (1 μL) of protein and reservoir solution. The crystallization conditions for *Vd*PDA1 (1.1 M sodium malonate, 0.1 M HEPES, 0.5% v/v Jeffamine ® ED-2001; pH 7.0) and Pst_13661 (1260 mM ammonium sulfate, 100 mM sodium cacodylate/hydrochloric acid; pH 6.5) were obtained from Index (Hampton Research) and Wizard 1 (Rigaku). The enzyme-ligand complex crystals were harvested using incubation proteins with a 10-fold molar excess of the compounds and crystalized under the same conditions (1260 mM ammonium sulfate, 100 mM sodium cacodylate/hydrochloric acid; pH 6.5), as described above.

Crystals were harvested in rayon fiber loops and flash-frozen in liquid nitrogen. The diffraction data were collected using 0.954 Å (*Vd*PDA1), 0.97852 Å (Pst_13661), 0.97925 Å (Pst_13661-BHA, Pst_13661-Compound **2** and Pst_13661-Compound **3**) at 100 K on Beamline BL18U1 (*Vd*PDA1), BL19U1 (Pst_13661), and BL17B1 (Pst_13661-BHA, Pst_13661-Compound **2** and Pst_13661-Compound **3**) of the National Facility for Protein Science (NFPS) at the Shanghai Synchrotron Radiation Facility in China. The data were processed and scaled using the HKL3000 package[47].

### Structure determination and refinement

Data analyses were performed using Phenix. The structures of *Vd*PDA1 (PDB ID: 8HFA), Pst_13661 (PDB ID: 8HF9), and Pst_13661-BHA (PDB ID: 8HE1), Pst_13661-compound **2** (PDB ID: 8HE2), and Pst_13661-compound **3** complexes (PDB ID: 8HE4) were resolved by molecular replacement with Phaser[48] using the structure of *Cl*CDA (PDB ID 2IW0) as a model. The PHENIX program suite was used for structure refining[49]. Ramachandran statistics showed 96.57% (*Vd*PDA1), 96.4 % (Pst_13661), 97.3% (Pst_13661-BHA), 96.9% (Pst_13661-compound **2**) and 97.3% (Pst_13661-compound **3**) of residues in the most favored region

**Table 2 | Inhibitory activity of BHA and its derivatives against *Vd*PDA1 and Pst_13661**

| Compound | Structure | VdPDA1 | | Pst_13661 | |
|---|---|---|---|---|---|
| | | Inhibition rate (%, mean ± SD, 100 μM) | $K_i$ (μM) | Inhibition Rate (%, mean ± SD, 100 μM) | $K_i$ (μM) |
| 1 | | 96.3 ± 0.4 | 8.31 | 100 ± 5.5 | 9.83 |
| 2 | | 93.4 ± 2.9 | 8.53 | 100 ± 2.7 | 10.75 |
| 3 | | 89.5 ± 1.3 | 9.63 | 85.1 ± 0.9 | 27.58 |
| 4 | | 76.8 ± 3.1 | 27.59 | 66.4 ± 3.6 | 80.68 |
| 5 | | 59.6 ± 8.5 | -* | 30.3 ± 2.2 | - |
| 6 | | 27.8 ± 4.2 | - | 34.4 ± 6.8 | - |
| 7 | | 1.8 ± 4.5 | - | 23.5 ± 5.4 | - |
| 8 | | 0.0 | - | 18.1 ± 8.6 | - |
| 9 | | 0.0 | - | 8.2 ± 8.4 | - |

*Not determined.

and 3.43% (*Vd*PDA1), 3.6% (Pst_13661), 2.7% (Pst_13661-BHA), 3.1% (Pst_13661-compound **2**) and 2.7% (Pst_13661-compound **3**) in the allowed regions. Refinement values are given in Table 1. Coot was used to manually build and extend molecular models[50]. The quality of the models was checked with PROCHECK[51]. The structural figures were created using the molecular visualization software PyMol[52] and UCSF ChimeraX[53]. The diffraction data and structure refinement statistics are summarized in Table 1.

To identify which metal ion was coordinated by the Asp-His-His triad in the active pocket, X-ray fluorescence spectra scan of a Pst_13661 crystal was carried on BL18U1 Beamline of NFPS (Shanghai, China)[54]. The data were analyzed using the software OriginPro. Zn-SAD of Pst_13661 were then performed. For zinc searching and phasing, the Pst_13661 structure (8HF9), with the metal ion removed, was put into phaser MR-SAD in the PHENIX program suite against the Zn-SAD data of Pst_13661, using the method described by Zhou et al.[55].

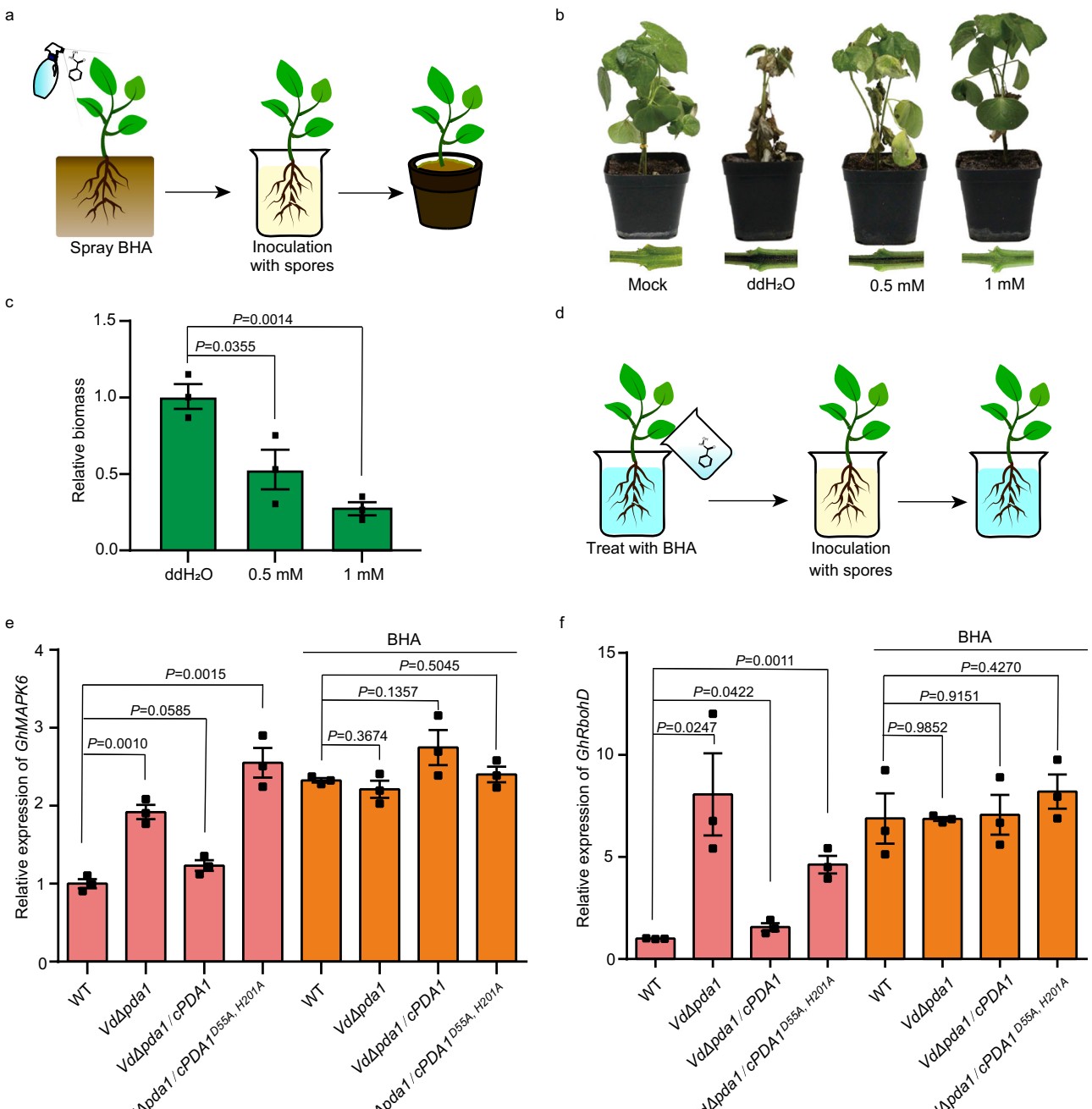

**Fig. 3 | Anti-*V. dahliae* activity of BHA relies on its inhibition of *Vd*PDA1.**
**a** Scheme of the anti-virulence analysis methodology. **b** Disease phenotypes and vascular discoloration of the stems and tissues of cotton seedlings. ddH$_2$O was used as the solvent for BHA. **c** Fungal biomass, as determined by qRT-PCR, on cotton plants infected by wild-type *Vd*PDA1 with BHA pre-treatment. **d** Scheme of the root-dip methodology. **e, f** Pre-treatment of cotton with BHA restores the expression of *GhMPK6* and *GhRbohD* that was suppressed by VdPAD1. The expression patterns of *GhMPK6* (**e**) and *GhRbohD* (**f**) in cotton roots at 36 hpi with the wild-type (WT), the

knockout mutant (Vd$\Delta$pda1), and the complementary transformants (Vd$\Delta$pda1/cPDA1$^{D55A, H201A}$, Vd$\Delta$pda1/cPDA1) were determined by RT-qPCR using cDNA as a template. Three independent replicates were used for measurement, and each experiment consisted of 3 cotton samples per *V. dahliae* infection. Bars in **c, e,** and **f** represent mean values, whiskers represent mean ± SEM, and black circles represent each replicate value ($n = 3$). Statistical significance was determined by Student's two-sided unpaired $t$ test. Source data are provided as a Source Data file.

## Enzymatic activity assay

The enzymatic activity of recombinantly expressed proteins was assayed using a fluorogenic labeling method[29]. The reaction mixture consisted of 100 µL of 100 nM testing enzyme and 0.5 mM (GlcNAc)$_3$ in a 50 mM Hepes (pH 7.5) buffer. After incubation for 10 min at 30 °C, free amine production was measured using fluorescamine labeling and quantified with a glucosamine standard. All measurements were carried out in triplicate. Source data are provided as a Source Data file.

A metal-dependent activity assay was performed according to the following steps: The protein was extensively dialyzed against 20 mM dipicolinic acid (DPA) in 50 mM Hepes (pH 7.5) buffer (3 buffer changes, 6 h each), followed by dialysis against 50 mM Hepes (pH 7.5) buffer to remove the DPA. The reaction was run for 10 min with 0.5 mM (GlcNAc)$_3$ and 100 nM of proteins. All metals were added as chloride salts at a concentration of 1 mM. All measurements were carried out in triplicate. Source data are provided as a Source Data file.

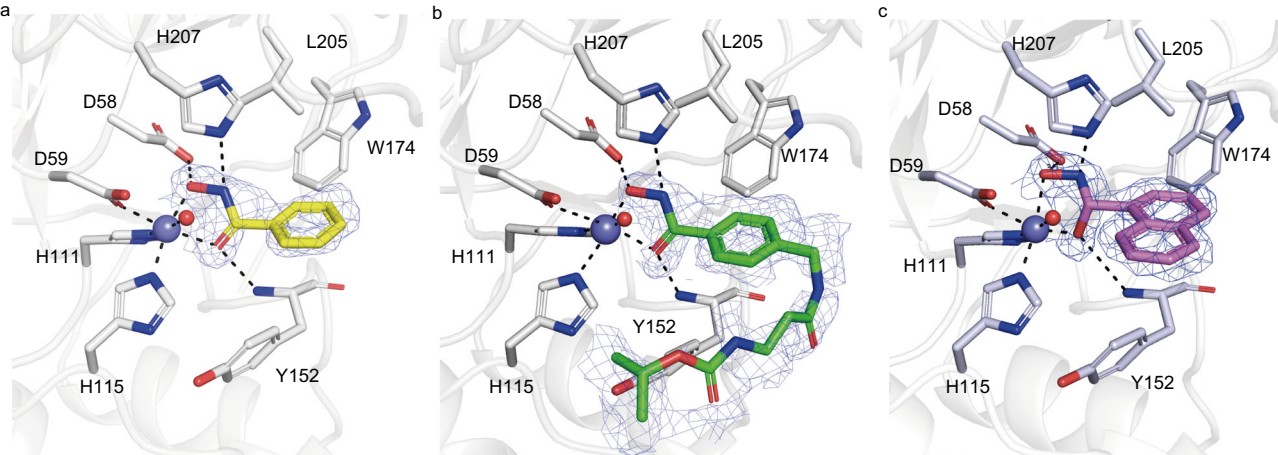

**Fig. 4 | Interactions of BHA and its derivatives with Pst_13661. a–c** Amino acid residues involved in the binding of BHA (**a**), compound **2** (**b**), and compound **3** (**c**) within the active site of Pst_13661. BHA, compound **2**, and compound **3** are shown as sticks with yellow, green, and light pink carbon atoms, respectively. The 2Fo-Fc electron-density map around the ligands is contoured at the 1.0 σ level. The Pst_13661 residues that interact with these compounds are shown as sticks with blue carbon atoms. Hydrogen bonds are shown as dashed lines. The zinc ions and water molecules are shown as slate and red spheres, respectively. The amino acids are labeled by their one-letter abbreviations.

## Screening of inhibitors and $K_i$ determination

The nine compounds were purchased from Topscience (http://www.tsbiochem.com), a local distributor of compounds from chemical vendors, including Chembridge, Chemdiv, and VitasM. The compounds were screened to identify the small molecules inhibiting VdPDA1 and Pst_13661 using a liquid-handling robot. The compounds were prepared in a 96-well plate containing one compound per well at 10 mM each in 100% DMSO (except for BHA, which was dissolved in ddH₂O). The plates were first diluted to 5 mM. The compounds were screened at 100 μM in 2% DMSO in a final reaction volume of 100 μL, containing 100 nM enzyme and 0.5 mM (GlcNAc)₃ in 50 mM Hepes (pH 7.5) buffer. Negative enzyme-free controls and positive controls with and without DMSO were prepared. The plates were incubated at 30 °C for 10 min. All measurements were carried out in triplicate. Source data are provided as a Source Data file.

For the determination of the inhibition constants $K_i$, three substrate (GlcNAc)₃ concentrations (0.5, 0.2, and 0.1 mM) and varied inhibitor concentrations were used. The $K_i$ values were determined using Dixon plots[56]. In Dixon plots, the value of lines intersection that converges above the x-axis is denoted as $K_i$. Dixon plots were obtained via curve fitting using the software GraphPad Prism. Source data are provided as a Source Data file.

## Phylogenetic analysis and sequence alignment

ClustalW (www.ebi.ac.uk/clustalw/) was used to perform multiple sequence alignments prior to phylogenetic analysis. MEGA 3.0[57] was used to construct the consensus phylogenetic tree using the maximum parsimony method and a cut-off value of 50% similarity. To evaluate the branch strength of the phylogenetic tree, a bootstrap analysis of 5000 replications was performed. The accession numbers of the fungal CE4 enzymes are shown in Supplementary Table 1.

Amino acid sequences of VdPDA1 from V. dahliae, Pst_13661 from P. striiformis f. sp. tritici, FoPDA1 from F. oxysporum f. sp. vasinfectum, FgCDA from F. graminearum, and RsCDA from R. solani were aligned using ClustalW.

## Construction of V. dahliae VdPDA1 mutants

Upstream and downstream genomic sequences of VdPDA1 were cloned and inserted into positions flanking the hygromycin resistance cassette of the pC1300-HPT vector to generate gene-knockout plasmids. To generate gene complementation plasmids for VdΔpda1/cPDA1 and VdΔpda1/PDA1^DSSA, H201A, the genomic sequences of VdPDA1 or VdPDA1^DSSA, H201A, including the 1.5 kb promoter region, were inserted into the pC1300G-mcherryHA₂ vector. Agrobacterium-mediated transformation-based gene-deletion strategy was used to generate Vdpda1-related transformants. Agrobacterium strains AGL-1 (Catalog number AC1020, Weidi, Shanghai, China) was used in the transformation of V. dahliae strain V991[58]. All transformants were selected using potato dextrose agar (PDA) medium with 50 mg/L hygromycin B for the knockout mutants or 40 mg/L G418 for the gene complementation mutants. The vectors were generously provided by Prof. Liu from the School of Food Science, Nanjing XiaoZhuang University, China. All oligonucleotide primers used in this study were summarized in Supplementary Table 2.

## Fungal growth at different concentrations of BHA

In this study, we used V. dahliae strain V991 (named WT in this study and gifted by Prof. Tingli Liu from the School of Food Science, Nanjing XiaoZhuang University), F. graminearum strain Fg-1 (isolated from soybeans in China), F. oxysporum strain Fo-2 (isolated from soybeans in Shandong, China), R. solani strain RS-1 (isolated from soybeans in Jiangsu, China), and T. harzianum T27 (gifted by professor Xiliang Jiang from the Institute of Plant Protection, Chinese Academy of Agricultural Sciences). All fungal strains were stored on sterilized filter papers at −20 °C. The strains were reactivated before use by growing them on PDA plates at 25°C in the dark. Colony diameters on the PDA plates were measured after 2 d for T. harzianum and R. solani, 3.5 d for F. graminearum and F. oxysporum, and 4 d for V. dahliae. Agar plugs containing different concentrations of BHA or the corresponding solvent were placed in the center of the PDA plates. Representative photographs were taken after the corresponding time of culture at 25°C.

## Evaluation of the in vivo anti-virulence activity of BHA

V. dahliae was cultured on PDA plate for 1 week. Transfer one plate (90 mm) of mycelia by a sterilized blade into 100 ml Czapek-Dox medium with shaking at 200 rpm for 3 days at 25 °C in the dark to obtain conidia. Cotton (Gossypium hirsutum cv. Lvmian) was grown at 25°C in a greenhouse (14 h:10 h day:night photoperiod). To evaluate the anti-virulence activity of BHA on cotton, the 2-week-old cotton seedlings were pre-treated with BHA or ddH₂O (the solvent for BHA) and inoculated with conidia using the root-dip method[59]. To measure fungal biomass, the infected cotton plants were harvested at 30 dpi and flash-frozen in liquid nitrogen for DNA extraction using a

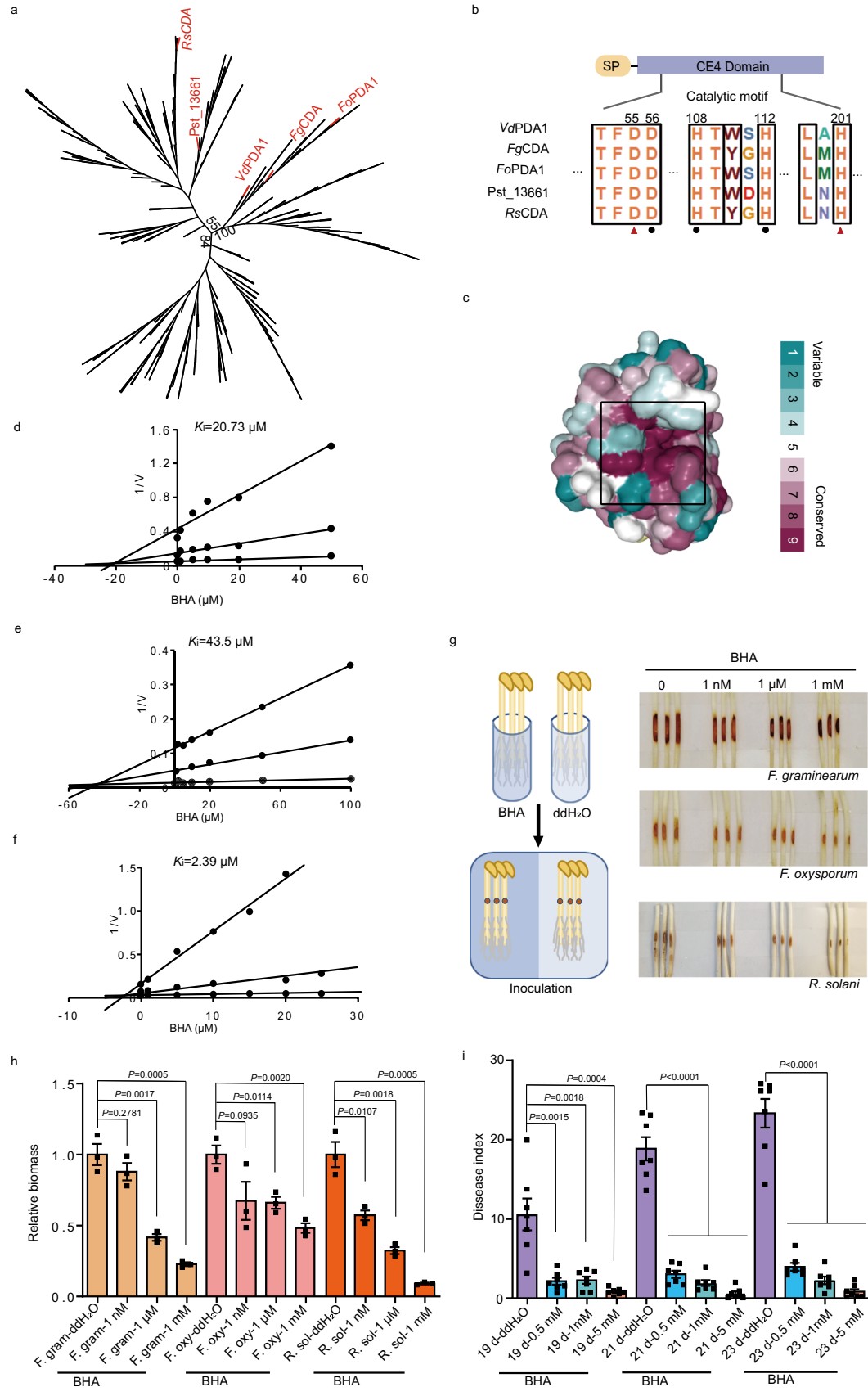

**Fig. 5 | Anti-virulence analysis of BHA against *F. graminearum, F. oxysporum, R. solani*, and *P. striiformis* f. sp. *tritici*. a** Phylogenetic analysis of pathogenic fungal CE4 deacetylases. **b** Conservation of the catalytic domains of *Vd*PDA1, *Fg*CDA, *Fo*PDA1, Pst_13661, and *Rs*CDA. SP represents the signal peptide. **c** Highly conserved residues around the active site of *Vd*PDA1, based on ConSurf analysis of 247 phytopathogenic fungal CDA sequences. The surface is colored by the ConSurf score according to the indicated scoring scheme. **d–f** Inhibition constant $K_i$ values of BHA against *Fg*CDA (**d**), *Fo*PDA1 (**e**), and *Rs*CDA (**f**). **g** Scheme of the anti-virulence analysis method and disease phenotypes of soybean hypocotyls infected with the indicated pathogens. **h** Fungal biomass, as determined by qRT-PCR, of soybean hypocotyls infected with the indicated pathogens with BHA or ddH$_2$O (the solvent) pre-treatment. Bars represent mean ± SEM ($n = 3$). Statistical significance was determined by Student's two-sided unpaired $t$ test. Source data are provided as a Source Data file. **i** Disease indices of wheat plants infected with *P. striiformis* f. sp. *tritici* at 19, 21, and 23 dpi, which were pre-treated with varying doses of BHA. Bars represent mean ± SEM ($n = 7$). Statistical significance was determined by Student's two-sided unpaired $t$-test. Source data are provided as a Source Data file.

cetyltrimethylammonium bromide (CTAB) assay. The fungal biomass in infected cotton plants was estimated using the value of the fungal-specific DNA relative to cotton actin, as determined by qRT-PCR. The qRT-PCR was performed in triplicate, and was repeated for three times. Primer sequences are listed in Supplementary Table 2. Source data are provided as a Source Data file.

To evaluate the anti-virulence activity of BHA on soybeans, the virulent fungal phenotypes were determined by inoculating etiolated soybean seedling hypocotyls, which were pre-treated with BHA or the solvent ddH$_2$O. To measure fungal biomass, the hypocotyl tissues of infected soybean plants were harvested at 48 hpi and flash-frozen in liquid nitrogen for DNA extraction using a CTAB assay. Virulence was determined using qPCR to measure the ratio of fungal to soybean DNA in the infected tissues at 48 hpi. The qRT-PCR was performed in triplicate, and was repeated for three times. Primer sequences are listed in Supplementary Table 2. Source data are provided as a Source Data file.

To evaluate the anti-virulence activity of BHA on the wheat cultivar Ming Xian 169 (ZM009379, http://icgr.caas.net.cn), parent wheat seedlings, which are highly susceptible to all known *P. striiformis* f. sp. *tritici* strains in China[60], were grown in a rust-free greenhouse. At the two-leaf stage, the plants were pre-treated with BHA or the solvent ddH$_2$O, and then uniformly inoculated with a single-spore of isolated *P. striiformis* f. sp. *tritici* (strain CYR32) after 24 h. Following inoculation, the phenotypes were recorded, and the disease index[61] was evaluated at 19, 21, and 23 dpi. All assays were repeated seven times. Source data are provided as a Source Data file.

### *GhMPK6* and *GhRbohD* gene expression measurements in plants
To measure *GhMPK6* and *GhRbohD* gene expression in plants, cotton seedlings were grown on plates with MS$^{1/2}$ medium (Murashige and Skoog, 1962 medium with 1/2 nutrients), for 5 d. We immersed the roots of cotton seedlings with conidial suspension (~1 × 10$^7$) for 1 h and carefully placed the inoculated seedlings back into the MS$^{1/2}$ medium. Roots were harvested and flash-frozen in liquid nitrogen at 36 hpi. Total RNA was extracted from cotton plant roots inoculated with the VdPDA1 or *VdPDA1* strains at 36 hpi using a Total RNA Purification Kit (Omega). One microgram of total RNA was used for cDNA synthesis using the HiScript IIQRT SuperMix for qPCR kit (Vazyme). *GhMPK6* and *GhRbohD* expression levels were measured using a 7500 real-time system (ABI) with SYBR Green PCR master mix (Vazyme). The qRT-PCR procedure consisted of 95°C for 30 s, 40 cycles of 95°C for 5 s, and 60°C for 34 s. The dissociation curves of each reaction were determined, and all reactions were performed with three biological replicates. Cotton actin genes were used as an internal control to normalize the expression values. Source data are provided as a Source Data file.

### Statistics and reproducibility
The sample sizes were reasonable numbers for the statistical analysis using in this paper. Sample sizes for pathogenicity and defense response gene expression on plants were nine (three technical repeats with three biological repeats), and for colony diameter assays were four plates per difference concentration BHA treatment. The exact value of n representing the number of repeats in the experiments was described in the figure legends. No data have been excluded. The findings of all key experiments were reliably reproduced. The experiment design and the experiment data collection are randomized. The infection assays were recorded in a blind way. Statistical analyses were performed using the software GraphPad Prism and Excel. $P$ values were determined by Student's unpaired two-sided $t$-test.

### Reporting summary
Further information on research design is available in the Nature Portfolio Reporting Summary linked to this article.

## Data availability
All data generated in this study are available in the main text, supplementary materials, or the source data file except for the structural data that have been deposited to the Protein Data Bank [https://www.rcsb.org] under the accession code 8HFA (*Vd*PDA1), 8HF9 (Pst_13661), 8HE1 (Pst_13661-BHA), 8HE2 (Pst_13661-compound **2**), 8HE4 (Pst_13661-compound **3**). The structure model for molecular replacement used in this study are available in the Protein Data Bank under accession code 2IW0. The sequences used in this study to generate phylogenetic tree are available in the NCBI database [https://www.ncbi.nlm.nih.gov/] under accession codes that are presented in Supplementary Table 1. Source data are provided with this paper.

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

## Acknowledgements

We thank Professor Daan van Aalten (University of Dundee) for discussions on CDA inhibitors. We thank the staff from BL18U1, BL19U1, and BL17B1 beamline of the National Facility for Protein Science in Shanghai (NFPS) at Shanghai Synchrotron Radiation Facility, for assistance during data collection. This work was supported by the National Natural Science Foundation of China (32001938 to L.L. and 31830076 to Q.Y.), the Shenzhen Science and Technology Program (KQTD20180411143628272 to Q.Y.), and the Special Funds for Science Technology Innovation and Industrial Development of Shenzhen Dapeng New District (PT202101-02 to Q.Y.).

## Author contributions

Q.Y. conceived and designed this project in consultation with Y.C.W., H.C., and Y.W.; Q.Y. and L.L. provided funding; L.L. and Y.L. conducted the structural and biochemical experiments; Q.Y., L.L., and Y.Z. analyzed the structural data; Y.X. conducted the construction of *V. dahliae* VdPDA1 mutants and the anti-virulence assays of BHA; X.S. and W.L. conducted the anti-virulence assay of BHA on cotton seedlings; X.Y. conducted the anti-virulence analysis of BHA on wheat plants; Q.Y., L.L., Y.X., and Y.L. wrote the manuscript.

## Competing interests

The authors declare no competing interests.
