## [Peer Review File · Nature Communications]

Reviewers' Comments:

Reviewer #1:

Remarks to the Author:

This paper systematically analyzed the importance of secreted chitin deacetylase (CDA) and its inhibitor Benzohydroxamic acid (BHA) in fungal infection plant, through protein expression and purification, crystallization and data collection, structure determination and refinement, enzyme activity assay, inhibitors screening, sequence analysis, mutant construction, inhibitor effect on fungal growth, inhibitor anti-virulence assessment, plant GhMAPK6 gene expression analysis etc. It revealed the spatial crystal structure of CDA and found that *Verticillium dahliae* relied on the CDA activity for full virulence, BHA is inhibitor of CDA, as an ion chelator, and demonstrated broad-spectrum anti-disease activity. The results of this study revealed the importance of CDA for the virulence of plant pathogens, and demonstrated that a possible route for controlling crop disease by using the inhibitors of CDA.

Generally, the data are described in a logic way and the manuscript is well written. However, to improve quality and clarity of this submission, there are some important points which need to take into consideration.

Comments to Author:

Some of problems are needed to be addressed.

1. The "Abstract" does not briefly address the contents of this study.
2. In Page 2, Line 15, it should be "benzohydroxamic acid (BHA) substituent".
3. In Page 2, Line 17, this study did not involve rice as the study material, but rather soybean was used.
4. In Page 4, Line 2-3, the reference 21 was erroneously cited. It should be powdery mildew, not be stem rust.
5. In Page 6, Line 11-16, Although the ChMPK6 gene is a defense response gene in the host plant, why was it only selected as the response gene for the CDA mutant in this study? Do other defense response genes have similar results?
6. In Page 6, Line 20-23, "Benzohydroxamic acid (BHA) is known to be a chelator that can chelate a variety of metal ions^{35,36}. Furthermore, BHA and its derivatives have been studied as inhibitors of several zinc- and nickel-containing metalloenzymes, such as urease and histone deacetylase inhibitors³⁷⁻⁴⁰." These sentences are best placed in the "discussion".
7. In Page 7, Line 6, Fig 3A, whether the control experiments with only spray solvent were tested at the same time?
8. In Page 7, Line 11, Which tissue in cotton was used for analysis the expression level of ChMPK6? root or leaf segment?
9. In Page 8, Line 18-20, The Fig S2 was in vitro test, yet Fig 5E and 5F were in vivo test. They cannot be described in a single sentence. Fig. S2 lacks a quantitative description and obvious symptoms.
10. In Page 9, Line 1, The "discussion" was poor. It should not be a repetition of the results.
11. In Page 14, Line 12, 20 and Page 15 Line 4, "without BHA", why not be treated with solvent?
12. In Page 14, Line 17, 24, How many times were repeated in qRT-PCR?

Reviewer #2:

Remarks to the Author:

This appears to be an interesting study, which is very well executed and nicely presented (high quality figures in particular). Although I have no in-depth expertise on the protein structure biology, the structures of fungal secreted chitin deacetylases (CDA) from the two plant-infecting fungal species (*Verticillium dahliae* and *Puccinia striiformis* f. sp. *tritici*) obtained in this study, seem to be reasonable and resemble the structures of CDA obtained from other fungi (non-pathogenic on plants) in previous studies by others. Using a structure-guided modelling, the authors then identified benzohydroxamic acid (BHA) as the lead inhibitory compound for these fungal CDAs and demonstrated that application of BHA to plants prior to fungal inoculation significantly reduced the fungal biomass. Moreover, this study revealed that the CDA activity is essential for fungal virulence. My main critique is that it may not be possible to utilise BHA as a fungicide as its effects may not be specifically targeted against plant pathogenic fungi, and will

probably also target beneficial fungi too, which is not desirable. BHA is a well-known metal chelator of metal ions and an inhibitor of many metalloprotein enzymes such as histone deacetylases, matrix metalloproteases, etc and thus I also question relative novelty of this discovery. I also think that the title of this article gives the message that is a little too strong as this study showed that BHA attenuates fungal diseases of plants rather than "prevent crop diseases". I also wonder whether attenuation of diseases seen following BHA treatments in this study is actually due to specific inhibitory action towards CDA or is perhaps also due to inhibition of many more metalloproteins.

Minor comments:

- 1) Page 2, L11 – "Tritici" should not be capitalised.
- 2) Page 4, L2-3 – *Blumeria graminis* is the causal agent of powdery mildew rather than wheat stem rust.
- 3) Page 6, L21 – should read "ion" rather than "iron".
- 4) Page 7, L18-19 – I think it cannot be completely ruled in these experiments that the observed effects are specifically due to BHA targeting of CDA in vivo.

Reviewer #3:

Remarks to the Author:

The paper entitled "Inhibition of chitin deacetylases to prevent crop diseases" is an interesting study for the search for potent and promising CDA inhibitors to be used as anti-plant disease agents, based on BHA whose binding mode and antifungal properties have been determined. The article is well-written which makes it an interesting and comprehensive read. My review has focused on the use of protein crystallography, structure and kinetics contained in this paper.

Protein crystallography and structure

Although authors claim it, the conditions for obtaining crystals of apo and complexed forms are not the same since the former were obtained at pH 6.5 (line 250) and the latter at pH 6.8 (line 254). Is it a mistake?

Have authors calculate OMIT maps to verify the presence of ligands in the corresponding crystal structure? In figure 4, electron density map around ligands would be desirable to ensure the confidence in the ligand location.

Regarding Table 1.

It appears that there is a mistake in the numbered of compounds between Tables 1 and 2. Same compound is referred to differently in both tables: in Table 1, compound 4 (Table 1) is compound 2 in Table 2, and compound 2 (Table 1) is compound 3 in Table 2.

Value for $I/\sigma(I)$ for 8HE4 must be checked since the number for highest resolution shell is higher than the general one.

There is no legend to indicate what "a", "b" and brackets stand for.

For good models, one typically sees root-mean-square deviations (rmsd) of bond lengths less than 0.02 Å, and rmsd bond angles less than 2°. In 8HFA, rmsd deviation for bond length is 0.99 (an error??). Besides, the corresponding 8HFA_validation_report is NOT for manuscript review.

Regarding Figure 1S

Two peaks are displayed for Ni²⁺ and Zn²⁺. Is it possible that Ni²⁺ is present in the structures? For proteins containing a His-tag that were purified using Ni²⁺ affinity chromatography (most commonly used nickel-nitrilotriacetic acid Ni-NTA), the Ni²⁺ can be sequestered from an affinity column during purification and "contaminate" the sample. Then, this metal can bind and replace Zn²⁺. The authors should check this point and determine what Ni²⁺ peak means.

Kinetics:

The assays were performed correctly and the data collected from them was processed correctly. Nevertheless, despite the details of the enzyme activity assay being reported previously, the fitting equation for K_{i} determination should be included (line 290).

Reviewer #1 (Remarks to the Author):

This paper systematically analyzed the importance of secreted chitin deacetylase (CDA) and its inhibitor Benzohydroxamic acid (BHA) in fungal infection plant, through protein expression and purification, crystallization and data collection, structure determination and refinement, enzyme activity assay, inhibitors screening, sequence analysis, mutant construction, inhibitor effect on fungal growth, inhibitor anti-virulence assessment, plant GhMAPK6 gene expression analysis etc. It revealed the spatial crystal structure of CDA and found that *Verticillium dahliae* relied on the CDA activity for full virulence, BHA is inhibitor of CDA, as an ion chelator, and demonstrated broad-spectrum anti-disease activity. The results of this study revealed the importance of CDA for the virulence of plant pathogens, and demonstrated that a possible route for controlling crop disease by using the inhibitors of CDA.

Generally, the data are described in a logic way and the manuscript is well written. However, to improve quality and clarity of this submission, there are some important points which need to take into consideration.

Response: We are grateful to the reviewer for the time and effort taken in reviewing this manuscript and for providing constructive suggestions that have helped us improve the paper.

Comments to Author:

Some of problems are needed to be addressed.

1. The “Abstract” does not briefly address the contents of this study.

Response: We have curtailed the “Abstract” to 150 words to briefly address the contents of this study.

2. In Page 2, Line 15, it should be “benzohydroxamic acid (BHA) substituent”.

Response: We have corrected the mistake (Page 2, line 34).

3. In Page 2, Line 17, this study did not involve rice as the study material, but rather soybean was used.

Response: We have removed “rice” as the study material (Page 2, line 36 and Page 4, line 90).

4. In Page 4, Line 2-3, the reference 21 was erroneously cited. It should be powdery mildew, not be stem rust.

Response: We have corrected “wheat stem rust” to “powdery mildew” (Page 3, line 68).

5. In Page 6, Line 11-16, Although the ChMPK6 gene is a defense response gene in the host plant, why was it only selected as the response gene for the CDA mutant in this study? Do other defense response genes have similar results?

Response: We thank the reviewer for the questions. The mitogen-activated protein kinase 6 (*MPK6*) that we chose for this study and the respiratory burst oxidase homolog protein D (*RbohD*), are the two chitin-induced defense response genes previously known for cotton roots (Gao F, et al. *Nature plants* 2019, 5:1167-1176; Huang W, et al. *International journal of molecular sciences* 2021, 22; Bozsoki Z, et al. *Proceedings of the National Academy of Sciences of the United States of America* 2017, 114: E8118-E8127; Kaku H, et al. *Proceedings of the National Academy of Sciences of the United States of America* 2006, 103:11086-11091.). In response to your questions, we investigated the other defense response gene, *GhRbohD*, for the

study. The results indicated that the expression pattern of *GhRbohD* was similar to that of *GhMPK6* when cotton plants were infected by pathogens carrying either active or inactive CDAs (please see the figures attached below). The detailed experiments associated with *GhRbohD* and the corresponding results (Fig. 2e-g & Fig. 3e, f) have been added to the revised manuscript.

Fig. 2 (Page 25-26, lines 672–694)

e **Enzymatic activity is essential for the full virulence function of VdPAD1.** The fungal relative biomass on cotton plants at 30 dpi with the WT, the knock-out mutant (*VdΔpda1*), and the complementary transformants (*VdΔpda1/cPDA1*^{D55A, H201A}, *VdΔpda1/cPDA1*) was determined by qPCR, using genomic DNA as a template. Bars in the graphs represent mean values, whiskers represent mean ± SEM, and black circles represent each replicate value (n = 3). Different letters (a-b) represent significant differences from a low to high level (Duncan's multiple range test; $P < 0.01$).

f, g **Expression of the defense response genes (*GhMPK6* and *GhRbohD*) that was repressed by VdPAD1 during infection by *V. dahliae*.** The expression patterns of *GhMPK6* (f) and *GhRbohD* (g) in cotton roots at 36 hpi with the WT, the knock-out mutant (*VdΔpda1*), and the complementary transformants (*VdΔpda1/cPDA1*^{D55A, H201A}, *VdΔpda1/cPDA1*) were determined

by qRT-PCR using cDNA as a template. Three independent replicates were used for measurement, and each experiment consisted of 3 cotton samples per *V. dahliae* infection. Bars in the graphs represent mean values, whiskers represent mean \pm SEM, and black circles represent each replicate value (n = 3). Different letters (a-b) represent significant differences from a low to high level (Duncan's multiple range test; $P < 0.01$).

Fig. 3 (Page 27, lines 696–711)

e, f Pre-treatment of cotton with BHA restores the expression of GhMPK6 and GhRbohD that was suppressed by VdPAD1. The expression patterns of *GhMPK6* (D) and *GhRbohD* (E) in cotton roots at 36 hpi with the WT, the knock-out mutant (*VdApda1*), and the complementary transformants (*VdApda1/cPDA1*^{D55A, H201A}, *VdApda1/cPDA1*) were determined by RT-qPCR using cDNA as a template. Three independent replicates were used for measurement, and each experiment consisted of 3 cotton samples per *V. dahliae* infection. Bars in all graphs represent mean values, whiskers represent mean \pm SEM, and black circles represent each replicate value (n = 3). Different letters (a-d) represent significant differences from a low to high level (Duncan's multiple range test; $P < 0.01$).

6. In Page 6, Line 20-23, “Benzohydroxamic acid (BHA) is known to be a chelator that can chelate a variety of metal irons^{35,36}. Furthermore, BHA and its derivatives have been studied

as inhibitors of several zinc- and nickel-containing metalloenzymes, such as urease and histone deacetylase inhibitors³⁷⁻⁴⁰." These sentences are best placed in the "discussion".

Response: These sentences have been moved and modified to the Discussion section (Page 9, lines 209–211).

7. In Page 7, Line 6, Fig 3A, whether the control experiments with only spray solvent were tested at the same time?

Response: Thanks for the question. The control experiments with only the spray solvent were tested at the same time. Here, the spray solvent was ddH₂O because BHA is water soluble. We added the solvent information to the Methods section (Page 15, line 350).

8. In Page 7, Line 11, Which tissue in cotton was used for analysis the expression level of ChMPK6? root or leaf segment?

Response: Thanks for the questions. Cotton root was used for analyzing the expression level of *GhMPK6* and *GhRbohD*. The analysis was based on a study by Gao F, et al. *Nature plants* 2019, 5(11):1167-1176. We have added the detailed information in the revised manuscript (Page 7 line 156 and Page 16 line 376).

9. In Page 8, Line 18-20, The Fig S2 was in vitro test, yet Fig 5E and 5F were in vivo test. They cannot be described in a single sentence. Fig. S2 lacks a quantitative description and obvious symptoms.

Response: We have described Fig. S2 (Fig. S3 in the revised manuscript) and Fig. 5E-F in different sentences (Page 8, line 187-191), and we have added quantitative data and symptoms in Fig. S3 (Page 35, line 767-773, please also see the figures below).

Supplementary Figure 3. Effects of BHA on fungal mycelial growth. **a** Colonies of *R. solani*, *F. oxysporum*, *F. graminearum*, *V. dahliae*, and *T. harzianum* on PDA plates. PDA plates were dipped with BHA at different concentrations. **b** Diameter of fungal colonies. The diameter was measured when the fungi grew to about two-thirds of the PDA plate. All experiments were performed in four repeats. The letter “a” represents no significant difference according to Duncan’s multiple range test ($P < 0.01$).

10. In Page 9, Line 1, The "discussion" was poor. It should not be a repetition of the results.

Response: Thanks for the comment and suggestion. We have removed the repetition of the results. In addition, we have added a section on the possible limitations of using BHA, as it might interact with other metalloproteins and not only CDAs (Page 9, line 204-219 and Page 10, line 220-222).

11. In Page 14, Line 12, 20 and Page 15 Line 4, “without BHA”, why not be treated with solvent?

Response: We apologize for the ambiguous wording. We have corrected “without BHA” to “with the solvent ddH₂O”. We used ddH₂O as the solvent because BHA is water-soluble (Page 15, line 350 and 358).

12. In Page 14, Line 17, 24, How many times were repeated in qRT-PCR?

Response: The qRT-PCR was performed in triplicate, and was repeated for three times. We have added this information to the revised manuscript (Page 15, line 355; Page 16, line 362).

Reviewer #2 (Remarks to the Author):

This appears to be an interesting study, which is very well executed and nicely presented (high quality figures in particular). Although I have no in-depth expertise on the protein structure biology, the structures of fungal secreted chitin deacetylases (CDA) from the two plant-infecting fungal species (*Verticillium dahliae* and *Puccinia striiformis* f. sp. *tritici*) obtained in this study, seem to be reasonable and resemble the structures of CDA obtained from other fungi (non-pathogenic on plants) in previous studies by others. Using a structure-guided modelling, the authors then identified benzohydroxamic acid (BHA) as the lead inhibitory compound for these fungal CDAs and demonstrated that application of BHA to plants prior to fungal inoculation significantly reduced the fungal biomass. Moreover, this study revealed that the CDA activity is essential for fungal virulence.

Response: We are very thankful for the reviewer's comments and grateful for the time taken in reviewing this manuscript.

My main critique is that it may not be possible to utilize BHA as a fungicide as its effects may not be specifically targeted against plant pathogenic fungi, and will probably also target beneficial fungi too, which is not desirable.

Response: Thanks for the thoughtful comment. Our results indicated that BHA had no effects on the growth of pathogenic fungi, including *F. oxysporum*, *F. graminearum*, and *R. solani* (Supplementary Fig. 3), as well as the bio-agent fungus *Trichoderma harzianum* (please see our newly performed experiments, Supplementary Fig. 3, Page 35, line 767-773). In this scenario, we tend to conclude that BHA was more likely a plant immune activator and might not target

beneficial fungi. However, there might be other side effects of BHA application in crop protection. Further investigations on BHA's limitations need to be performed.

BHA is a well-known metal chelator of metal ions and an inhibitor of many metalloprotein enzymes such as histone deacetylases, matrix metalloproteases, etc and thus I also question relative novelty of this discovery.

Response: Thanks for the thoughtful comment. We believe that the novelty of our discovery is that compounds bearing a BHA core can inhibit pathogenic fungal chitin deacetylases. Until now, it was not known that BHA could inhibit chitin deacetylase, an enzyme that uses chitin as its substrate. Recently, we found that BHA and its derivatives could not inhibit chitin deacetylases derived from insects (Supplementary Fig. 4, please see the data below), indicating that BHA has a preference to fungal chitin deacetylases, even though it was previously thought to be a broad-spectrum inhibitor of deacetylases. This discovery will undoubtedly lead to the rational design of inhibitors that target pathogenic fungal chitin deacetylases, based on the BHA scaffold. This notion was previously unknown to both plant protection researchers and agrochemical designers.

Supplementary Figure 4. Effect of BHA on the activity of *BmCDA7* from the insect *Bombyx mori*. Bars in all graphs represent mean values, whiskers represent mean \pm SEM, and black

circles represent each replicate value ($n = 3$). The letter “a” represents no significant difference according to Duncan’s multiple range test ($P < 0.01$).

I also think that the title of this article gives the message that is a little too strong as this study showed that BHA attenuates fungal diseases of plants rather than “prevent crop diseases”.

Response: The title has been revised to “Inhibition of chitin deacetylases to attenuate plant fungal diseases” (Page 1, line 1).

I also wonder whether attenuation of diseases seen following BHA treatments in this study is actually due to specific inhibitory action towards CDA or is perhaps also due to inhibition of many more metalloproteins.

Response: BHA has the potential to inhibit other metalloproteins, although it is not yet known if there are any other pathogenic fungal metalloproteins involved in immune escape from host plants’ defensive responses. Our data only establishes a direct link between CDA and BHA. As a CDA inhibitor, BHA did not affect the growth of either the host plant or the fungus, but it did attenuate fungal diseases. Further investigations are needed to determine if other mechanisms are involved, or if they can be ruled out.

Minor comments:

1) Page 2, L11 – “Tritici” should not be capitalised.

Response: "Tritici" has been corrected to "tritici" (Page 2, line 31).

2) Page 4, L2-3 – *Blumeria graminis* is the causal agent of powdery mildew rather than wheat

stem rust.

Response: The phrase "wheat stem rust" has been corrected to "powdery mildew". (Page 3, line 68).

3) Page 6, L21 – should read “ion” rather than “iron”.

Response: The word “iron” has been corrected to “ion”.

4) Page 7, L18-19 – I think it cannot be completely ruled in these experiments that the observed effects are specifically due to BHA targeting of CDA *in vivo*.

Response: We have revised the sentence and put it in the following way: “these results suggest that BHA might inhibit CDA *in vivo*” (Page 7, line 163-164).

Reviewer #3 (Remarks to the Author):

The paper entitled "Inhibition of chitin deacetylases to prevent crop diseases" is an interesting study for the search for potent and promising CDA inhibitors to be used as anti-plant disease agents, based on BHA whose binding mode and antifungal properties have been determined. The article is well-written which makes it an interesting and comprehensive read

Response: We are grateful to the reviewer for taking the time and effort to further improve this manuscript.

My review has focused on the use of protein crystallography, structure and kinetics contained in this paper.

Protein crystallography and structure

1. Although authors claim it, the conditions for obtaining crystals of apo and complexed forms are not the same since the former were obtained at pH 6.5 (line 250) and the latter at pH 6.8 (line 254). Is it a mistake?

Response: Thanks for pointing out the mistake. We have corrected the pH "6.8" to "6.5" (Page 11, line 259).

2. Have authors calculate OMIT maps to verify the presence of ligands in the corresponding crystal structure? In figure 4, electron density map around ligands would be desirable to ensure the confidence in the ligand location.

Response: We have calculated OMIT maps to verify the presence of ligands in the

corresponding crystal structures. Fig. 4 (Page 28, lines 712–720) has been redrawn to show the electron density map around the ligands (please see below).

Figure 4. Interactions of BHA and its derivatives with Pst_13661. **a-c** Amino acid residues involved in the binding of BHA (**a**), compound **2** (**b**), and compound **3** (**c**) within the active site of Pst_13661. BHA, compound **2**, and compound **3** are shown as sticks with yellow, green, and light pink carbon atoms, respectively. The 2Fo-Fc electron-density map around the ligands is contoured at the 1.0 σ level. The Pst_13661 residues that interact with these compounds are shown as light blue sticks. Hydrogen bonds are shown as dashed lines. The zinc ions and water molecules are shown as slate and red spheres, respectively.

Regarding Table 1.

3. It appears that there is a mistake in the numbered of compounds between Tables 1 and 2. Same compound is referred to differently in both tables: in Table 1, compound 4 (Table 1) is compound 2 in Table 2, and compound 2 (Table 1) is compound 3 in Table 2.

Response: The compounds have been renumbered to make sure they are consistent in Tables 1 and 2.

4. Value for $I/\sigma(I)$ for 8HE4 must be checked since the number for highest resolution shell is higher than the general one.

Response: Thanks for pointing out this mistake. We have corrected the value for $I/\sigma(I)$.

5. There is no legend to indicate what "a", "b" and brackets stand for.

Response: We have added the legends to indicate what "a", "b", and brackets stand for. The "a" represents " $R_{merge} = \frac{\sum hkl \sum j |I_j(hkl) - \langle I(hkl) \rangle|}{\sum hkl \sum j I_j(hkl)}$, where I is the intensity of reflection and $h, k,$ and l are the indices of the reflections". The "b" represents " $R_{pim} = \frac{\sum hkl [1/(N-1)]^{1/2} \sum j |I_j(hkl) - \langle I(hkl) \rangle|}{\sum hkl \sum j I_j(hkl)}$, where N is the redundancy of the dataset". The values in brackets represent high-resolution data.

6. For good models, one typically sees root-mean-square deviations (rmsd) of bond lengths less than 0.02 Å, and rmsd bond angles less than 2°. In 8HFA, rmsd deviation for bond length is 0.99 (an error???)

Response: Thanks a lot for pointing out the mistake. The values for bond lengths and angles were misplaced. We have corrected the error.

7. Besides, the corresponding 8HFA_validation_report is NOT for manuscript review.

Response: We have provided the 8HFA_validation_report.

Regarding Figure S1

8. Two peaks are displayed for Ni^{2+} and Zn^{2+} . Is it possible that Ni^{2+} is present in the structures?

For proteins containing a His-tag that were purified using Ni^{2+} affinity chromatography (most

commonly used nickel-nitrilotriacetic acid Ni-NTA), the Ni^{2+} can be sequestered from an affinity column during purification and “contaminate” the sample. Then, this metal can bind and replace Zn^{2+} . The authors should check this point and determine what Ni^{2+} peak means.

Response: Thank you very much for the question and comment. To identify which metal ion is coordinated by the Asp-His-His triad in the active pocket, we performed zinc single-wavelength anomalous dispersion (SAD) of Pst_13661. For zinc searching and phasing, we used the method described by Zhou et al. *Nature Communications* 2016, 7, 13082. In brief, the previous Pst_13661 structure (8HF9), with the metal ion removed, was put into phaser MR-SAD in the PHENIX program suite against Zn-SAD data of Pst_13661. The position of zinc was determined to be around the Asp-His-His triad (Supplementary Figure 1b, please also see the figure below). In this scenario, Ni^{2+} was likely to be a contaminant sequestered from nickel-nitrilotriacetic acid (Ni-NTA) affinity chromatography during purification.

Supplementary Figure 1. b Stereo views of anomalous difference density for Zn^{2+} in the structure of Pst_13661. The $2\text{Fo}-\text{Fc}$ map around Zn^{2+} is contoured at the 2.0 level. (Page 33, line 749-759).

We rechecked the structure of Pst_13661 and did not observe the density of a divalent metal ion at the C-terminus, where the His-tag is located. This might be because a 15 residue-long sequence at the C-terminus could not be resolved. Besides structural biology data, we obtained biochemical data that indicate that both Pst_13661 and *VdPDA1* prefer Zn^{2+} for their activity

(Supplementary Figure 1c,d, please also see the figure below).

Supplementary Figure 1 c, d Relative activity of Pst_13661 (c) and VdPDA1 (d) with Zn²⁺ and Ni²⁺. After treatment with 20 mM dipicolinic acid (DPA) to remove the original metal ion, the protein was incubated with 1 mM metal cation (ZnCl₂, NiCl₂), and the activity was determined with 0.5 mM (GlcNAc)₃ as the substrate. Bars in all graphs represent mean values, whiskers represent mean ± SEM, and black circles represent each replicate value (n = 3). Different letters (a-d) represent significant differences from a low to high level (Duncan's multiple range test; P < 0.01). (Page 33, line 749-759).

Taken together, both Pst_13661 and VdPDA1 carry a Zn²⁺ ion for their activity. The aforementioned information has been added to the revised manuscript.

Kinetics:

9. The assays were performed correctly and the data collected from them was processed correctly. Nevertheless, despite the details of the enzyme activity assay being reported previously, the fitting equation for K_i determination should be included (line 290).

Response: Thank you for your suggestion. The fitting equations for K_i determination have been provided in the Fig.5d-f and Supplementary Fig. 2 (please also see the figures below).

Figure 5 d-f Inhibition constant K_i values of BHA against *FgCDA* (**d**), *FoPDA1* (**e**), and

RsCDA (**f**). (Page 29, line 722-730).

Supplementary Figure 2. Inhibition constant K_i values of BHA derivatives against *VdPDA1* and *Pst_13661*. **a-d** The determination of K_i values of compound 1 (a), compound 2 (b), compound 3 (c), compound 4 (d) toward *VdPDA1*; **e-h** The determination of K_i values of compound 1 (e), compound 2 (f), compound 3 (g), compound 4 (h) toward *Pst_13661*. (Page 34, line 761-765).

Reviewers' Comments:

Reviewer #1:

Remarks to the Author:

I note that the authors have point-to-point responded the raised questions and the revised or supplemented parts were displayed with highlighted bright bars in the revised version. I think these modifications are acceptable.

Reviewer #2:

Remarks to the Author:

I am happy with the authors' responses to the reviewers' comments and the corresponding changes that they have made to the manuscript. My advice now would be to accept the manuscript for publication.

Reviewer #3:

Remarks to the Author:

I am pleased to report that almost all of my comments have been addressed by the authors in this second revision. For that, they have also conducted supplemental figures and data processing.

Nevertheless, the authors may have misunderstood my last comment on kinetics.

The assays were performed correctly and the data collected from them was processed correctly. Nevertheless, despite the details of the enzyme activity assay being reported previously, the fitting equation for K_i determination should be included (line 290).

I cannot see the point in providing the fitting equations on each line in all corresponding figures. (Fig 5 d,e,f and Supplementary Figure 2) as authors have done in response to that comment. I meant rather, that a generic fitting equation should be added in Methods, indicating the meaning of the values where line crosses each axe.

Reviewer #3 (Remarks to the Author):

I am pleased to report that almost all of my comments have been addressed by the authors in this second revision. For that, they have also conducted supplemental figures and data processing.

Response: We are grateful to the reviewer for the time and effort in reviewing this manuscript and providing valuable suggestions that have helped us improve this manuscript.

Nevertheless, the authors may have misunderstood my last comment on kinetics. I cannot see the point in providing the fitting equations on each line in all corresponding figures. (Fig 5 d,e,f and Supplementary Figure 2) as authors have done in response to that comment. I meant rather, that a generic fitting equation should be added in Methods, indicating the meaning of the values where line crosses each axe.

Response: We have removed the fitting equations on each line in all corresponding figures. (Fig. 5 d,e,f and Supplementary Figure 2). Alternatively, we have added a description in Methods to indicate the meaning of the values where line crosses each axe.

The added description is as follows: “For the determination of the inhibition constants K_i , three substrate (GlcNAc)₃ concentrations (0.5, 0.2, and 0.1 mM) and varied inhibitor concentrations were used. The K_i values were determined using Dixon plots (Dixon M. *Biochemical journal* 1953, 55(1): 170.). In Dixon plots, the value of lines intersection that converges above the x-axis denoted K_i . Dixon plots were obtained via curve fitting using the software GraphPad Prism.”.